# The political consequences of opioid overdoses

**Aaron R. Kaufman** [1]*, **Eitan D. Hersh**[2]

**1** Division of Social Sciences, New York University Abu Dhabi, Saadiyat Island, Abu Dhabi, United Arab Emirates, **2** Department of Political Science, Tufts University, Medford, Massachusetts, United States of America

* aaronkaufman@nyu.edu

**Data Availability Statement:** Data are available from https://doi.org/10.7910/DVN/SO1CBG.

**Funding:** The authors received no specific funding for this work.

## Abstract

The United States suffered a dramatic and well-documented increase in drug-related deaths from 2000 to 2018, primarily driven by prescription and non-prescription opioids, and concentrated in white and working-class areas. A growing body of research focuses on the causes, both medical and social, of this opioid crisis, but little work as yet on its larger ramifications. Using novel public records of accidental opioid deaths linked to behavioral political outcomes, we present causal analyses showing that opioid overdoses have significant political ramifications. Those close to opioid victims vote at lower rates than those less affected by the crisis, even compared to demographically-similar friends and family of other unexpected deaths. Moreover, among those friends and family affected by opioids, Republicans are 25% more likely to defect from the party than the statewide average Republican, while Democrats are no more likely to defect; Independents are moderately more likely to register as Democrats. These results illustrate an important research design for inferring the effects of tragic events and speak to the broad social and political consequences of what is becoming the largest public health crisis in modern United States history.

## The opioid crisis in the United States

The opioid crisis is among the most serious public health crises the US has ever faced [1]: more than 30,000 people per year have died of opioid overdoses since 2014, the vast majority between 25 and 54 years of age. Increases in opioid-related deaths in the United States from 2000 to 2015 have contributed to as much of a loss of life expectancy (0.28 years) as car accidents, suicide, Alzheimer's, liver failure, and septicemia combined (0.33)[2], and now account for more than 10% of accidental deaths in the US [3]. Its rapid spread and devastating death toll has rendered the opioid epidemic an issue of high political salience: in 2019 the Center for Disease Control allocated $475 million to track and mitigate its spread, and the National Institutes of Health launched an initiative in 2017 to reverse overdoses, treat addition, and remedy pain non-addictively [4].

The opioid epidemic is a problem rooted in a social, economic, and political context [5, 6]. Some blame a corporate ecosystem of pharmaceutical companies and unscrupulous doctors

**Competing interests:** The authors have declared
that no competing interests exist.

for over-marketing and over-prescribing highly addictive opioids [7], while others point to social and economic determinants like poverty, unemployment, and limited economic mobility as important moderating factors of drug use and abuse. [8–10].

While the public health effects of the opioid epidemic are clear, the social and political effects are largely unstudied, though a study from the beginning of the crisis estimates economic costs of opioid users as $13,000 Canadian dollars per year [11]. A broad literature in political science examines the consequences of traumatic events like terrorism [12], race riots [13], and war [14], largely finding politically mobilizing effects; there is good reason to suspect that the opioid crisis has produced complex political aftershocks as well. And while the effects of such shocks to individuals may be pronounced, they may also serve as drivers of local or national policy, as victims of such tragic events often become spokespeople in advocating for large policy changes.

This paper applies a validated causal research design to study the political consequences of the opioid epidemic on the individuals closest to it—the friends and family of overdose victims. Leveraging recent computational advantages and newly available large-scale government databases, we show that the opioid epidemic has significantly altered patterns of party identification and turnout, but with important heterogeneity across partisanship. In contrast to other studies examining the effects of tragedy on political participation, we find that friends and family of opioid overdose victims are less likely to turn out to vote than they were before tragedy struck, even compared to victims of premature cancer or a demographically-matched sample of registrants without familial opioid overdoses. Moreover, opioid overdoses seem to change the beliefs and preferences of those affected: Independents are far more likely to re-register as Democrats, and Republicans are nearly 50% more likely to defect to other parties, but Democrats' party affiliations are unchanged. More broadly, we expand the literature on the persistent effects of shocks to political behavior by showing a context in which participation declines rather than increases, and where shifts in party identification are causally identified, offering new insights into possible long-term consequences of large public health crises.

## Hypotheses

How might friends and family of opioid overdose victims respond politically? We examine two sets of outcomes: political engagement and partisan preference. Prior research finds positive engagement effects for crime victimization [15], immigration [16], race riots [13] and terrorist attacks [12], so we have reason to suspect that those affected by the opioid epidemic may mobilize to affect policy change. Another line of research, however, suggests that sadness and depression might decrease voter mobilization [17], offering a contrasting hypothesis.

On partisan preferences, too, prior research is split: whereas victims of terrorism become on average more conservative as national identity becomes more important [12], those proximate to race riots become more liberal, perhaps as they become more aware of the struggles of racial minorities [13]. And in considering the psychological repercussions of opioid overdose, while anxiety may lead to more liberal preferences [18], fear may lead to more conservative preferences instead [19]. Our measure of partisan preference is the change in registered party affiliation, a strong behavioral indicator of preferences relative to survey reporting. Finally, a large literature observes that apolitical events like sports victories, hurricanes, and shark attacks tend to disadvantage the incumbent party, whom voters blame for misfortunes [20], suggesting that overdose victims' families may defect from the Democratic party during the Democratic administrations, or defect to it during Republican administrations; an equally large literature points out that events like overdoses may increase the salience of healthcare-

related issues in the minds of voters, pushing them toward the party with the most ownership over the healthcare debate [21, 22].

## Data & methodology

How do these conflicting hypotheses manifest in the friends and family of opioid overdose victims? Our analysis strategy leverages panel data with voter behavior linked to accidental drug death records. We identify in public voter registration records the friends and family of opioid overdose victims, and compare their election turnout rates and party identification in the elections immediately before and after their loved one's death. Opioid overdose victims are not a representative sample of the population, however, so any political differences we observe may be due to demographic differences instead. To avoid this inferential pitfall we identify a control group of individuals who prematurely died of cancer (which we define as 50 years old or younger, corresponding to approximately the youngest 10% of cancer deaths [23]), and perform coarsened exact matching with Mahalanobis distance [24] to obtain a sample of individuals who died of cancer that look demographically identical to our sample of individuals who died of opioid overdoses.

In this way, we causally identify the effect of having a loved one die of an opioid overdose as compared to another type of premature death. To threaten our causal identification, an unobserved variable would have to confound the difference between the change in party identification among opioid overdose victims' families and the change in turnout or party identification among cancer victims' families. (Figs 1 and 2). This comparison separates out the effect of premature deaths from the effects of an opioid overdose death in particular, isolating the bundle of treatments distinctive to opioid deaths such as politicization, media attention, and stigmatization.

### Voter files

We begin with a panel of two Connecticut voter files: a 2013 voter file with 2,421,897 individuals and a 2017 voter file with 2,428,165 individuals; Connecticut uses a unique voter ID which we use to match registrants from 2013 to 2017, producing 2,001,250 matched individuals. The Connecticut voter file contains party identification information as well as election turnout for as more than a decade of previous federal, statewide, and local elections. We merge these data with geography-based census measures from the National Historical Geographic Information System (NHGIS, see Table 1).

### Overdose file

The Office of the Chief Medical Examiner in Connecticut makes free available a de-identified list of accidental opioid-related deaths; our version of this data contains 3,583 records. These records contain the location of the deceased individual, their age at death, their race, sex, town of residence, and the opioid(s) identified in their toxicology reports. We limit our analyses to those overdoses that occurred after the 2012 general election but before the 2016 primary election.

### Obituary records

To match these individuals to the voter file, we first re-identify them by searching for their dates, locations, and ages at death in publicly searchable obituary records. We take all uniquely identified opioid overdose obituaries and manually record their full names and precise birth dates. This results in 1,369 opioid deaths in our data set. We also identify the friends and family

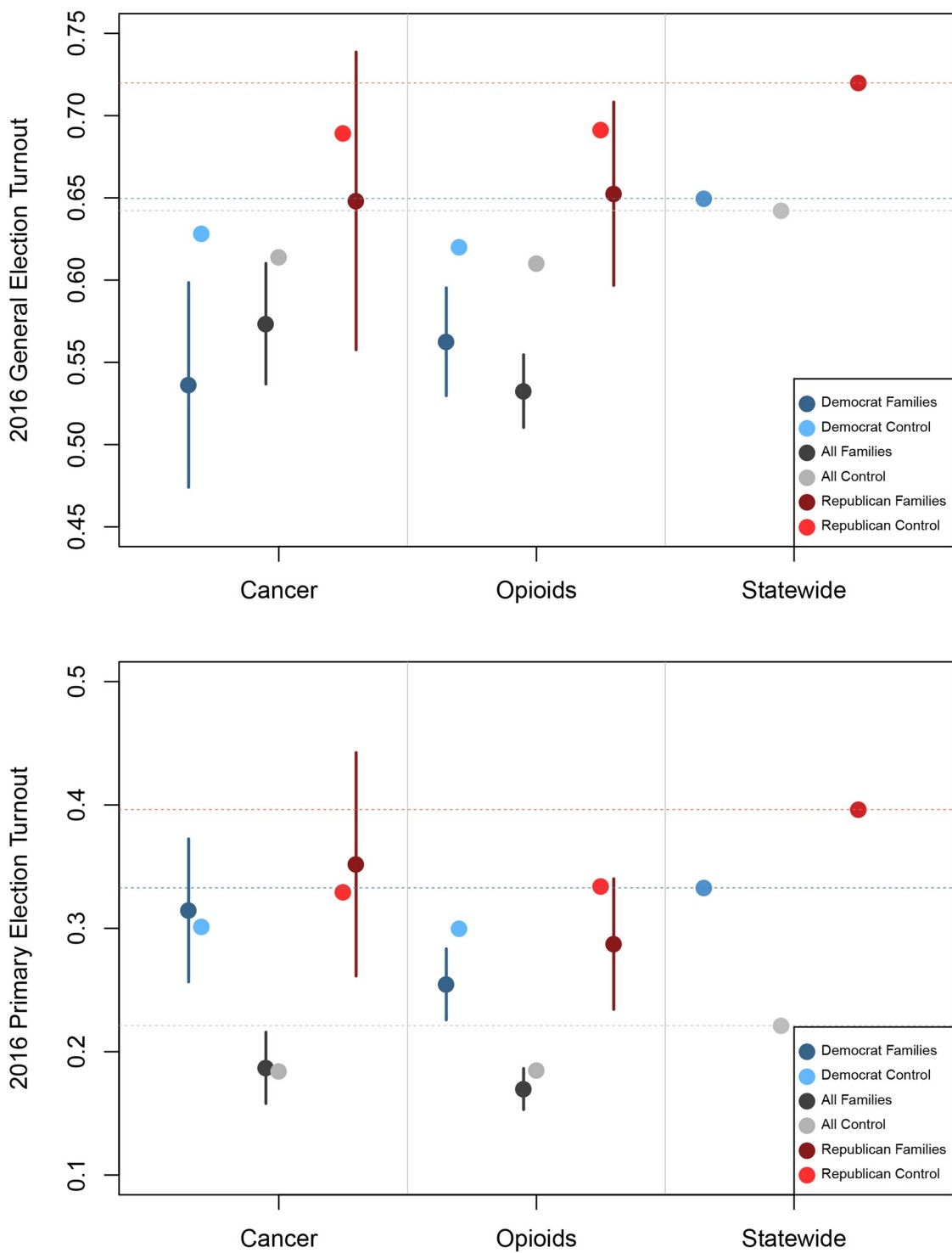

**Fig 1. Voter turnout among registered voters in the 2016 Presidential election (top) and the 2016 primary election (bottom), among the families of cancer victims, opioid overdose victims, and among all voters.** Blue (red) dots indicate turnout among Democrats (Republicans) only; grey dots indicate overall means (Republicans, Democrats, and Independents). Darker dots indicate turnout for among the relevant category (cancer families or opioid overdose families), while lighter dots indicate turnout among a matched sample from the voter file. In the 2016 Presidential election (top panel), registered Democrats who are the family members of a cancer victim voted at a rate of 54%, 9 percentage points lower than a matched sample whose family did not suffer a cancer death.

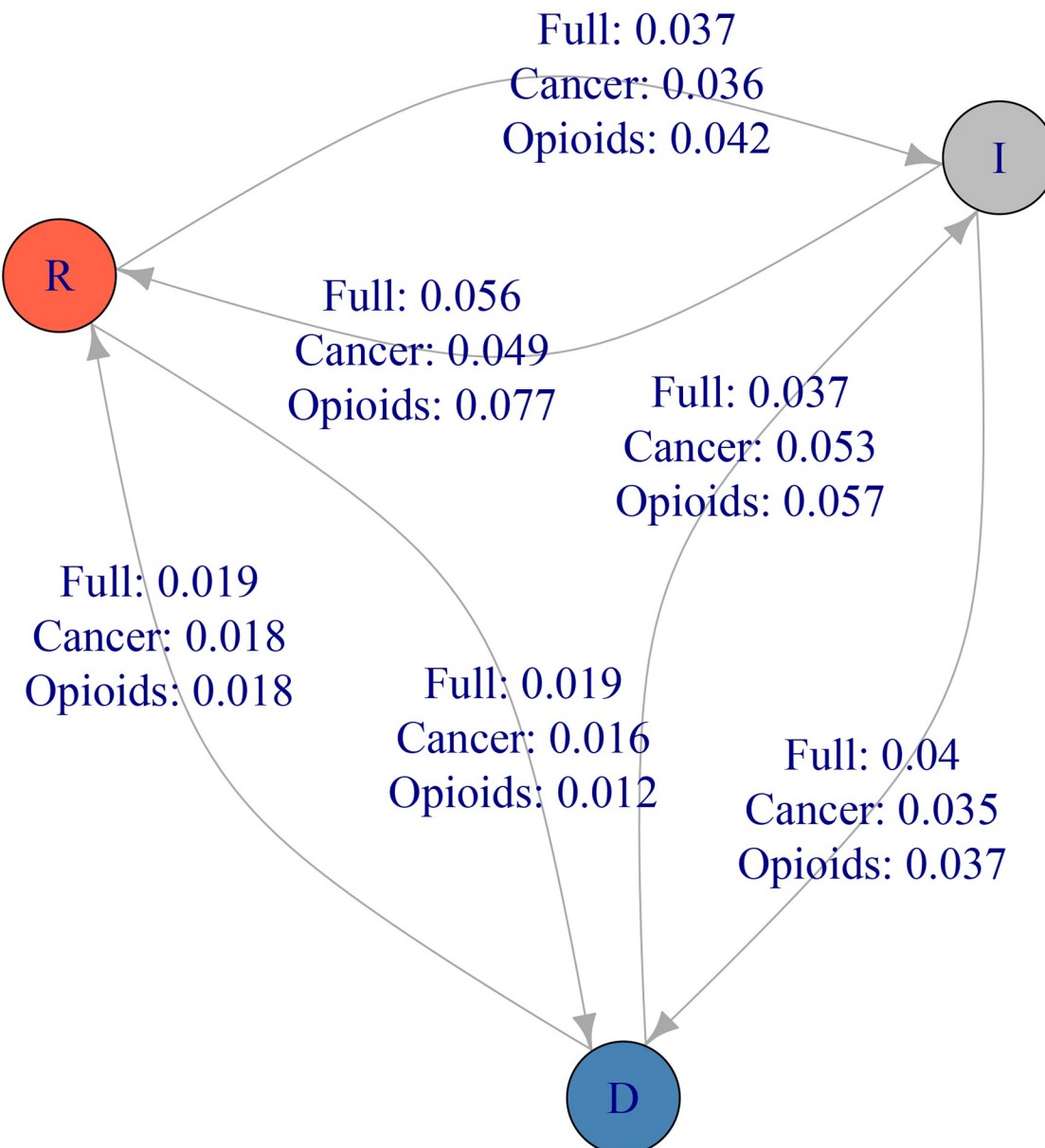

**Fig 2. A party affiliation transition matrix from 2012 to 2016.** Democrats who suffered an opioid overdose or a cancer death in the family are more likely to register as an Independent than the average Democrat.

of these overdose victims by locating those individuals living at the same address as the overdose victim at the time of death, and find 1,972 such individuals. We exclude cases with more than 10 registered voters living at the same address as they may not share a friend or family relationship with the victim.

## Cancer records

For our comparison sample, we search obituary records for premature cancer deaths that occurred after the 2012 general election but before the 2016 primary election, then manually

**Table 1. Descriptive statistics.** OD victims and their families are both more Democratic and less likely to vote than either the state average or cancer victims and their families. Note that 2012 turnout is conditional on registering to vote.

|  | CT Pop | Cancer | | Overdose | |
|---|---|---|---|---|---|
|  |  | Deaths | Families | Deaths | Families |
| Male | 0.416 | 0.427 | 0.379 | 0.544 | 0.418 |
| White | 0.872 | 0.909 | 0.928 | 0.865 | 0.811 |
| Black | 0.024 | 0.055 | 0.021 | 0.038 | 0.052 |
| Hispanic | 0.084 | 0.036 | 0.030 | 0.091 | 0.130 |
| Asian | 0.020 | 0.000 | 0.021 | 0.005 | 0.007 |
| Age | 52.182 | 47.518 | 55.267 | 48.375 | 48.081 |
| % Democrat | 0.365 | 0.400 | 0.350 | 0.438 | 0.448 |
| Pres. Turnout 2012 | 0.582 | 0.445 | 0.603 | 0.422 | 0.563 |
| N | 2,421,897 | 110 | 705 | 1,369 | 1,972 |

record their full names and precise birth dates. We find 110 such deaths and, using the same procedure as for the overdose victims, we find 705 household members.

## Matching

Using coarsened exact matching (CEM) with Mahalanobis distance, we identify two samples of matched voters: one that looks demographically similar to the family members of cancer victims, and another that looks demographically similar to the family members of opioid overdose victims. By comparing political behaviors of these matched groups to the original samples of family members, we control for all observed (but not unobserved) factors that might confound the relationship between cancer or opioid overdose deaths and voter turnout. The variables in our matching procedure include race, age, sex, county, household occupancy, census block median household income, and prior turnout history.

We summarize our data in Table 1 below. Overdose deaths disproportionately fall on Hispanics and African-Americans, on men rather than women, and on the young more than the old, but otherwise resemble the population relatively well. Unsurprisingly, cancer victims also have substantially more household members than opioid victims. However, we do observe that victims of overdose deaths and their families vote at lower rates than the statewide average.

## Election turnout

We first examine voter turnout (Fig 1). The statewide turnout rate in the 2016 General Election in Connecticut was approximately 65% (top panel rightmost grey dot and dashed grey line); Republicans voted at a higher rate (dashed red line). The overall turnout rate among the families of cancer victims (dark grey dot) is about 8 percentage points lower from the overall turnout rate, while the rate among families of opioid victims is about 10 percentage points lower. Democrats among the families of both sets of victims vote at lower rates; both Democrats and Republicans among the families of both sets of victims vote approximately 10 percentage points less than their copartisans who were unaffected, and to a lesser degree, less than the demographically-matched samples of copartisans. A different pattern emerges for primary election turnout: the friends and family of opioid overdose victims vote at a rate about 4 percentage points lower than the overall rate, and both Democrat and Republican friends and family of opioid overdose victims vote at lower rates than the average, but cancer victim families of neither party vote at lower rates.

**Table 2. Rates of party change.** The statewide rate of leaving the Republican party is 5.6%, but among the loved ones of opioid overdose victims, that rate 25% higher. The Control columns indicate registered voters matched to cancer and overdose victims' families. Substantively important differences are bolded.

| | All CT | Cancer | | | Overdose | | | Diff-in-diff |
|---|---|---|---|---|---|---|---|---|
| | | Families | Control | Diff. | Families | Control | Diff. | |
| R → !R | 0.056 | 0.074 | 0.062 | **0.012** | 0.074 | 0.063 | **0.011** | -0.001 |
| D → !D | 0.056 | 0.052 | 0.061 | -0.009 | 0.053 | 0.060 | -0.007 | 0.002 |
| !R → R | 0.030 | 0.027 | 0.030 | -0.003 | 0.024 | 0.030 | -0.006 | -0.003 |
| !D → D | 0.044 | 0.042 | 0.051 | **-0.009** | 0.062 | 0.050 | **0.012** | **0.021** |
| R → R | 0.944 | 0.926 | 0.938 | -0.012 | 0.926 | 0.937 | -0.011 | 0.001 |
| R → I | 0.037 | 0.056 | 0.041 | **0.015** | 0.057 | 0.041 | **0.016** | 0.001 |
| R → D | 0.019 | 0.019 | 0.021 | -0.002 | 0.018 | 0.022 | -0.004 | -0.002 |
| I → R | 0.040 | 0.035 | 0.039 | -0.004 | 0.037 | 0.039 | -0.002 | 0.002 |
| I → I | 0.904 | 0.916 | 0.900 | **0.016** | 0.886 | 0.902 | **-0.016** | **-0.032** |
| I → D | 0.056 | 0.049 | 0.061 | **-0.012** | 0.077 | 0.059 | **0.018** | **0.030** |
| D → R | 0.019 | 0.016 | 0.019 | -0.003 | 0.012 | 0.019 | -0.007 | -0.004 |
| D → I | 0.037 | 0.036 | 0.042 | -0.006 | 0.041 | 0.040 | 0.001 | 0.007 |
| D → D | 0.944 | 0.948 | 0.939 | 0.009 | 0.947 | 0.940 | 0.007 | -0.002 |

## Party defection

Turning to party affiliation (Table 2), we examine the rates at which the friends and family of opioid overdose victims change parties relative to cancer victims' friends and family as well as the statewide average. Statewide, both Democrats and Republicans left their party from 2012 to 2016 at a rate of 5.6%; among Democrats, both cancer and opioid overdose victims' friends and family leave the party at approximately the same rate—5.2% and 5.3% compared to 5.6% statewide. Among Republicans, however, both cancer and opioid overdose victims' friends and family leave the party at much higher rates—both 7.4%—than the statewide average of 5.6%. In comparison, demographically-matched control Republicans left the party at rates of 6.2% and 6.3%, indicating a 1 percentage point estimated treatment effect of premature death on Republican party defection. This change is almost entirely due to Republicans becoming Independent (5.6% and 5.7%) rather than becoming Democrats (1.9% and 1.8%). Among Independents, the friends and family of cancer victims join a major party at about the state-wide rate, but Independent friends and family of opioid overdose victims join the Democratic Party at a rate 1/3 higher than the statewide rate. To test the significance of these differences, we conduct Chi-squared tests between pairs of columns represented as transition matrices. Chi-squared tests indicate a statistically significant difference ($p < 0.01$) between the statewide and OD families transition matrices and between the cancer families and OD families transition matrices, but not between the statewide and cancer families transition matrices. As well, the difference between the OD families and OD matched control transition matrices is significant, but the difference between the cancer families and cancer matched control transition matrices is not.

## Discussion

The opioid epidemic, much like COVID-19, represents a political conflict rooted in a public health crisis, and it is therefore important to examine its political ramifications. In this study we have shown that the opioid epidemic substantially affects the political behavior among the friends and family of its victims. While tragic deaths in general reduce turnout in the general election, only opioid epidemic deaths reduce turnout in the primary election compared to

other premature deaths. Moreover, turning to partisanship, Republican friends and family were increasingly likely to defect from the GOP, Independents were increasingly likely to join the Democratic party, both suggesting a liberalizing trend.

The shifting partisan allegiances suggest that, in the face of opioid-driven tragedy, voters do not appear to hold the Democratic party accountable for opioid overdoses; note that during this period of study, both the US presidency and the Connecticut governorship were held by Democrats. To the contrary, voters act as though they prefer more liberal policies toward managing the opioid crisis in the face of self-interest or socialization [25, 26], even if they may not know the differences between the parties' policies [27].

We do note third key limitations of our results. First, our analysis is restricted to a single, unrepresentative state. Secondly, we only find obituary records for 38.2% of opioid overdose victims in the state, and that sample may be unrepresentative of opioid overdose victims as a whole. Finally, by identifying cancer deaths through obituary records, we may have an unrepresentative sample of statewide cancer deaths. Importantly, for this unrepresentativeness to bias our internal validity, cancer victims with and without obituaries that specifically mention cancer would need to be systematically different in a manner correlated with changing party registration. Taken together, these three limitations restrict our ability to generalize our findings to other states and populations. However, if we could safely assume that the trends we observe in Connecticut extrapolate to the entire US, based on results from [28] we would estimate that the opioid overdose resulted in 200,000 fewer cast votes in the 2016 Presidential election, nearly 40,000 additional Democratic registrants, and an equal number of fewer Republican registrants merely among the friends and family of its victims (see S1 Text).

Extending this line of inquiry may involve interrogating whether this change in party affiliation is a policy-driven response to self-interest or a broader shift in outlook and emotional state in response to tragedy. Finally, our results add important nuance extant literature on the politically mobilizing effects of dramatic events like terrorism and race riots, and future research in this field may also examine and characterize which types of shocks are mobilizing and which depress turnout, or which shocks tend to liberalize or conservatize.

## Supporting information

**S1 Text.**
(PDF)

## Acknowledgments

We thank Sheng Ha and two anonymous reviewers for invaluable feedback. We are indebted to Neal Beck, Leo Celi, Brian Libgober, Luke Miratrix, Robert Ward, and Madeleine Wolf for helpful comments during the preparation of this manuscript, and to ObituaryData.com of generously providing access to national obituary data. All remaining errors are our own. This work was approved by the Harvard University Committee on the Use of Human Subjects, IRB Protocol 19-0015; it was designated "Not Human Subjects Research".

## Author Contributions

**Conceptualization:** Aaron R. Kaufman.

**Data curation:** Aaron R. Kaufman.

**Formal analysis:** Aaron R. Kaufman.

**Funding acquisition:** Aaron R. Kaufman.

**Investigation:** Aaron R. Kaufman.

**Methodology:** Aaron R. Kaufman, Eitan D. Hersh.

**Project administration:** Aaron R. Kaufman, Eitan D. Hersh.

**Resources:** Aaron R. Kaufman.

**Software:** Aaron R. Kaufman.

**Supervision:** Aaron R. Kaufman, Eitan D. Hersh.

**Validation:** Aaron R. Kaufman.

**Visualization:** Aaron R. Kaufman.

**Writing – original draft:** Aaron R. Kaufman.

**Writing – review & editing:** Aaron R. Kaufman, Eitan D. Hersh.

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
