## [Decision Letter · Decision Letter 0]

24 Jun 2020

PONE-D-20-14110

The political consequences of opioid overdoses

PLOS ONE

Dear Dr. Kaufman,

Thank you for submitting your manuscript to PLOS ONE. After careful consideration, we feel that it has merit but does not fully meet PLOS ONE’s publication criteria as it currently stands. Therefore, we invite you to submit a revised version of the manuscript that addresses the points raised during the review process.

We look forward to receiving your revised manuscript.

Kind regards,

Shang E. Ha, Ph.D.

Academic Editor

PLOS ONE

Journal Requirements:

2) Thank you for your ethics statement : "This study was approved as Harvard IRB protocol 19-0015."

3) We note that you have stated that you will provide repository information for your data at acceptance. Should your manuscript be accepted for publication, we will hold it until you provide the relevant accession numbers or DOIs necessary to access your data. If you wish to make changes to your Data Availability statement, please describe these changes in your cover letter and we will update your Data Availability statement to reflect the information you provide.

Reviewers' comments:

Reviewer's Responses to Questions

**Comments to the Author**

1. Is the manuscript technically sound, and do the data support the conclusions?

Reviewer #1: Partly

Reviewer #2: Yes

2. Has the statistical analysis been performed appropriately and rigorously? 

Reviewer #1: Yes

Reviewer #2: I Don't Know

3. Have the authors made all data underlying the findings in their manuscript fully available?

Reviewer #1: Yes

Reviewer #2: Yes

4. Is the manuscript presented in an intelligible fashion and written in standard English?

Reviewer #1: Yes

Reviewer #2: No

5. Review Comments to the Author

Reviewer #1: Thank you for the opportunity to review “The Political Consequences of Opioid Overdoses.” This is an intriguing manuscript with a clever research design. I have a number of concerns, however, before I can recommend publication.

First, I should applaud the authors for addressing the important subject of opioid overdose, which has recently become a deeply politicized issue. And the approach to this – identifying opioid victims through a database and then matching their families to a voter file -- is very crafty and well executed.

One concern I have is whether this research approach is considered ethical. Did an IRB approve of this? No, the names of the victims and their families are not publicized, but the cases in the public records on opioid deaths were no doubt anonymized to protect the families, and I wonder if functionally de-anonymizing them for research purposes is okay.

Second, while I appreciated the authors comparing accidental opioid overdose victims to cancer victims, I think this comparison has some problems. For one, opioid overdose has recently received a great deal of media attention as some sort of private tragedy that many families are dealing with, and people are urged to check in on friends and relatives who may be suffering from it and get them to help. Having a loved one die of an opioid overdose is thus sometimes stigmatized as a failure on the party of friends and family. People rarely if ever blame someone for losing a relative to cancer. Perhaps this is the very difference that the authors seek to investigate, but the authors may wish to specify what it is they believe is so different about opioid addiction that would make deaths have partisan consequences.

Relatedly, the approach to identifying cancer victims is very different than the approach to identifying opioid overdose victims. For the former, as I understand it, the authors searched through publicly available obituaries, while the latter drew a complete list of names from a database. Obituaries, however, will not necessarily list the cause of death. Thus the sample of cancer victims is likely biased in some way, based on the wishes of the family members writing the obituaries, although it is not clear whether those biases would correlate with race, age, income, or the political variables the authors are investigating.

Finally, I’m not sure if this is appropriate in a relatively brief research piece like this, but perhaps it would be useful to have some theoretical motivation for why having an opioid death in the family would lead a family to change their voting behavior, or to leave one party but not another.

I hope these comments are useful.

Reviewer #2: This is a fascinating piece of research. The authors tackle an interesting and important question and they use a clever design to get at it. I could see this design having applications to similar issues (e.g., victims of police violence, COVID-19 deaths, etc.). So, I am all for publishing this piece. The only thing holding me up is that I do not believe that the authors present the data in an intuitive way. I stared at Table 2 and Figure 2 for a long time, comparing it to the paragraph describing the results, and I must say, I don't see how the numbers in the table support the interpretation. To my eyes, the results look almost identical for Cancer families and OD families. I will submit that I may being dense, but I might not be the only one! Could the authors summarize these findings in a more intuitive way? And in a way that incorporates confidence intervals in the party-switching analysis. Also note that Table 1 does not fit on the page, cutting off valuable information. This looks like it was a compiling error. Likewise Figure 1 was hard to follow as well. I would suggest labeling the dots in a legend on the figure itself.

6. PLOS authors have the option to publish the peer review history of their article (what does this mean?). If published, this will include your full peer review and any attached files.

Reviewer #1: No

Reviewer #2: No

---

## [Author Response · Author response to Decision Letter 0]

28 Jun 2020

Reviewer 1:

“One concern I have is whether this research approach is considered ethical. Did an IRB approve of this? No, the names of the victims and their families are not publicized, but the cases in the public records on opioid deaths were no doubt anonymized to protect the families, and I wonder if functionally de-anonymizing them for research purposes is okay.”

We share the Reviewer’s concerns for human subjects protection. We acknowledge that our data linkage procedure poses a threat to individual anonymity of sensitive populations, and in consultation with the Harvard University IRB, we took considerable measures to ensure that our data storage and management exceeded standard practice. Notably, we de-identify all named individuals in our data sets and store the data on servers that Harvard University has designated as Level 3 security. To reflect this, we have added a footnote to our Data & Methodology section indicating that the project “...was approved by the Harvard University Committee on the Use of Human Subjects, IRB Protocol 19-0015; it was designated Not Human Subjects Research.”

“Second, while I appreciated the authors comparing accidental opioid overdose victims to cancer victims, I think this comparison has some problems. For one, opioid overdose has recently received a great deal of media attention as some sort of private tragedy that many families are dealing with, and people are urged to check in on friends and relatives who may be suffering from it and get them to help. Having a loved one die of an opioid overdose is thus sometimes stigmatized as a failure on the party of friends and family. People rarely if ever blame someone for losing a relative to cancer. Perhaps this is the very difference that the authors seek to investigate, but the authors may wish to specify what it is they believe is so different about opioid addiction that would make deaths have partisan consequences.”

We thank Reviewer 1 for succinctly stating our comparison – since opioid overdoses have received so much media attention, there is a “bundle of treatments” that are involved in an opioid overdose that are NOT involved in a premature cancer death. Certainly stigmatization is part of that bundle! We have clarified this point in our Data & Methodology section. We gratefully borrow some of Reviewer 1’s language and write: “This comparison separates out the effect of premature deaths from the effects of an opioid overdose death in particular, isolating the bundle of treatments distinctive to opioid deaths such as politicization, media attention, and stigmatization.”

“Relatedly, the approach to identifying cancer victims is very different than the approach to identifying opioid overdose victims. For the former, as I understand it, the authors searched through publicly available obituaries, while the latter drew a complete list of names from a database. Obituaries, however, will not necessarily list the cause of death. Thus the sample of cancer victims is likely biased in some way, based on the wishes of the family members writing the obituaries, although it is not clear whether those biases would correlate with race, age, income, or the political variables the authors are investigating.”

Reviewer 1 raises an important point. We have added a note about this potential limitation to our Discussion section, noting: “For this unrepresentativeness to bias our internal validity, cancer victims with and without obituaries that specifically mention cancer would need to be systematically different in a manner correlated with changing party registration.” Unfortunately, without a statewide representative sample of cancer deaths, we cannot check this claim without making additional untestable assumptions.

“Finally, I’m not sure if this is appropriate in a relatively brief research piece like this, but perhaps it would be useful to have some theoretical motivation for why having an opioid death in the family would lead a family to change their voting behavior, or to leave one party but not another.” 

We firmly believe in motivating our quantitative analyses with theoretical motivations. To this effect, we have drawn additional attention to our Hypotheses section, which lays out theoretically-grounded reasons to expect increased turnout, decreased turnout, liberal shifts, and conservative shifts in response to opioid overdoses. This section now begins: “How might friends and family of opioid overdose victims respond politically? We examine two sets of outcomes: political engagement and partisan preference.”

Reviewer 2: 

“This is a fascinating piece of research. The authors tackle an interesting and important question and they use a clever design to get at it. I could see this design having applications to similar issues (e.g., victims of police violence, COVID-19 deaths, etc.). So, I am all for publishing this piece.”

We thank the Reviewer for her or his very kind words.

“The only thing holding me up is that I do not believe that the authors present the data in an intuitive way. I stared at Table 2 and Figure 2 for a long time, comparing it to the paragraph describing the results, and I must say, I don't see how the numbers in the table support the interpretation. To my eyes, the results look almost identical for Cancer families and OD families. I will submit that I may being dense, but I might not be the only one! Could the authors summarize these findings in a more intuitive way? And in a way that incorporates confidence intervals in the party-switching analysis.”

We apologize to Reviewer 2 for Table 2; we believe Reviewer 2’s confusion arises from poorly-named columns. The important comparisons, as we now explain in a substantially-rewritten Party Defection section, are between Cancer Families and Cancer Matched, and between OD Families and OD Matched. Those matched columns do not indicate cancer families matched to OD families; they indicate either cancer or OD families matched to registered voters who are neither OD nor cancer families. 

The comparison between what we now call the Cancer Families and the Cancer Matched Control columns indicates the treatment effect of a premature cancer death; the comparison between the OD Families and OD Matched Control columns indicates the treatment effect of an opioid overdose. The difference-in-differences between those four columns compares the treatment effect of opioid overdoses to the treatment effect of premature cancer deaths.

We have reorganized Table 2 hierarchically, renamed its columns, and added columns of differences to clarify these points

“Also note that Table 1 does not fit on the page, cutting off valuable information. This looks like it was a compiling error.”

We are grateful to the Reviewer for catching this! We have converted this table to a hierarchical table for clarity; it is now substantially narrower and, we believe, much more readable.

“Likewise Figure 1 was hard to follow as well. I would suggest labeling the dots in a legend on the figure itself.”

We have added a legend to Figure 1, and agree with Reviewer 1 that it makes the figure easier to interpret.

---

## [Decision Letter · Decision Letter 1]

15 Jul 2020

The political consequences of opioid overdoses

PONE-D-20-14110R1

Dear Dr. Kaufman,

We’re pleased to inform you that your manuscript has been judged scientifically suitable for publication and will be formally accepted for publication once it meets all outstanding technical requirements.

Kind regards,

Shang E. Ha, Ph.D.

Academic Editor

PLOS ONE

Additional Editor Comments (optional):

Reviewers' comments:

Reviewer's Responses to Questions

**Comments to the Author**

1. If the authors have adequately addressed your comments raised in a previous round of review and you feel that this manuscript is now acceptable for publication, you may indicate that here to bypass the “Comments to the Author” section, enter your conflict of interest statement in the “Confidential to Editor” section, and submit your "Accept" recommendation.

Reviewer #1: All comments have been addressed

Reviewer #2: All comments have been addressed

2. Is the manuscript technically sound, and do the data support the conclusions?

Reviewer #1: Yes

Reviewer #2: Yes

3. Has the statistical analysis been performed appropriately and rigorously? 

Reviewer #1: Yes

Reviewer #2: Yes

4. Have the authors made all data underlying the findings in their manuscript fully available?

Reviewer #1: Yes

Reviewer #2: Yes

5. Is the manuscript presented in an intelligible fashion and written in standard English?

Reviewer #1: Yes

Reviewer #2: Yes

6. Review Comments to the Author

Reviewer #1: (No Response)

Reviewer #2: I thank the authors for responding thoughtfully to my review. Table 2 is now MUCH clearer, and it all makes sense now! This is a great paper.

7. PLOS authors have the option to publish the peer review history of their article (what does this mean?). If published, this will include your full peer review and any attached files.

Reviewer #1: No

Reviewer #2: No

---

## [Editor Report · Acceptance letter]

16 Jul 2020

PONE-D-20-14110R1 

The political consequences of opioid overdoses 

Dear Dr. Kaufman:

I'm pleased to inform you that your manuscript has been deemed suitable for publication in PLOS ONE. Congratulations! Your manuscript is now with our production department. 

Kind regards, 

on behalf of

Dr. Shang E. Ha 

Academic Editor

PLOS ONE